# The Impact of Social Isolation during the COVID-19 Pandemic on Physical and Mental Health: The Lived Experience of Adolescents with Obesity and Their Caregivers

**DOI:** 10.3390/ijerph18063026

**Published:** 2021-03-15

**Authors:** Giada Pietrabissa, Clarissa Volpi, Michela Bottacchi, Vanessa Bertuzzi, Anna Guerrini Usubini, Henriette Löffler-Stastka, Tamara Prevendar, Giada Rapelli, Roberto Cattivelli, Gianluca Castelnuovo, Enrico Molinari, Alessandro Sartorio

**Affiliations:** 1Istituto Auxologico Italiano IRCCS, Psychology Research Laboratory, 20145, Milan, Italy; giada.pietrabissa@unicatt.it (G.P.); c.volpi@auxologico.it (C.V.); m.bottacchi@auxologico.it (M.B.); anna.guerriniusubini@unicatt.it (A.G.U.); giada.rapelli@unicatt.it (G.R.); r.cattivelli@auxologico.it (R.C.); gianluca.castelnuovo@unicatt.it (G.C.); enrico.molinari@unicatt.it (E.M.); 2Department of Psychology, Catholic University of Milan, 20123 Milan, Italy; vanessa.bertuzzi@unicatt.it; 3Department of Psychoanalysis and Psychotherapy, Medical University of Vienna, 1090 Vienna, Austria; 4Psychology Study Programme, Sigmund Freud University Vienna—Ljubljana Branch, 1000 Ljubljana, Slovenia; tamara.prevendar@gmail.com; 5Istituto Auxologico Italiano IRCCS, Experimental Laboratory for Auxo-Endocrinological Research, 28824 Piancavallo (VB), Italy; sartorio@auxologico.it; 6Istituto Auxologico Italiano IRCCS, Division of Auxology, 28824 Piancavallo (VB), Italy

**Keywords:** obesity, adolescent, caregiver, lived experience, coronavirus disease, social isolation, self-management, interpretive phenomenology, content analysis, clinical psychology

## Abstract

Adolescence is a complex developmental phase, made more complex by obesity and the social isolation imposed by the COVID-19 pandemic. The literature related to the impact of social isolation on obesity self-management in adolescents is scant and inconsistent. This paper describes the phenomenon from the perspectives of a sample of adolescents with obesity enrolled in an inpatients’ multidisciplinary rehabilitation program for weight-loss and their caregivers, and its impact on different life domains. Individual semi-structured ad hoc interviews were conducted with 10 adolescent-caregiver dyads, and narratives were qualitatively investigated using an interpretative phenomenology approach to data. Twenty participants took part in the study. The major themes that emerged from this study fall into five basic categories: (1) COVID-19 as an opportunity to reconsider what makes a good life; (2) Persistence in life; (3) Empowering relationship; (4) Daily routine in quarantine; (5) Lives on hold. Understandings drawn from this study may assist health care professionals in providing holistic support, and guidance to adolescents with weight-related issues and their caregivers who experience social isolation during the COVID-19 pandemic.

## 1. Introduction

To flatten the curve of coronavirus disease 2019 (COVID-19) pandemic, many governments around the world have implemented unprecedentedly strict preventive measures, such as prolonged school closures and home confinement [1,2]. Although necessary to limit the spread of the virus, responses to the COVID-19 emergency have led to incomparable and precipitous changes in human behavior [3].

Indeed, social isolation resulted in increased sedentary lifestyle and screen time among youth [4,5]. Associated stressors include fears of infection, frustration, and boredom, inadequate information, lack of in-person contacts with classmates, friends, and teachers, lack of personal space at home, and family financial loss.

Recent studies have reported negative health outcomes related to social isolation [6] such as anxiety, worrying, irritability, depressive symptoms, and post-traumatic stress disorder symptoms—in 18.9% to 43.7% among youth in Asian, European, and American countries [7,8,9]. Needless to say, the impromptu nature of such lockdown has triggered obesity in adolescents [10,11,12] and has posed those with obesity to a greater risk of additional weight gain [13,14]. This worldwide phenomenon is of great significance to earn the appellation of covibesity [15]. In the United States, growth in the prevalence of dysfunctional eating habits was observed among youth with obesity during COVID-19 [16]. Further, a study conducted in Italy during three weeks of lockdown by Pietrobelli et al. on a sample of 41 children and adolescents with obesity revealed an increased consumption of fruits, chips, red meat, and sugary drinks, sleep duration (0.65 h/day) and screen time (4.85 h/day), while exercise was reduced by two and a half hours per week [17].

Obesity is a global epidemic with an estimated 38.2 million children-adolescents (5 to 19 years) being overweight or obese worldwide [18,19]. Notoriously, childhood—besides being a powerful predictor of obesity in adulthood—puts children at increased risk for developing numerous health problems later in life, including diabetes and heart diseases. Besides, early research suggests that obesity may also increase the susceptibility of youth to serious illnesses like COVID-19 [20,21].

In this scenario, parents are often the closest and best resource for children and adolescents. A supportive relationship with parents and peers can help to buffer the stressful experience of COVID-19-related social isolation. Parents are often important role models for eating behaviors and physical exercise aptitudes in children [22], and good parenting skills become particularly crucial during home confinement.

However, high levels of parental stress due to job loss, isolation, and unexpectedly working from home [23] might negatively influence parental practices [24]. Many studies have analyzed the central role of the family environment in school-age children with obesity [25] and found strong associations between higher parental stress and the decrease of action in support of their children’s lifestyle change [26].

To mitigate the impact of COVID-19 measures and to promote psychological well-being and weight management, there is a need for more meaningful knowledge on the impact of COVID-19-related social isolation on the lives of adolescents with obesity and their caregiver.

To this end, this study is aimed to identify the main patterns in the lived-experiences that a sample of adolescents with obesity and their caregiver made of COVID-19-related social isolation, and to describe attitudinal, psychological, and behavioral responses to social isolation.

## 2. Materials and Methods

An interpretative phenomenological research design [27] was used to investigate multiple life domains, and the lived experiences that the adolescent-caregiver dyads made of COVID-19-related social isolation.

This design was deemed appropriate for the aim of the present study since—by giving voice to both adolescents and their caregivers—it allows the recognition of their way to perceive life through experiences, values, and norms.

### 2.1. Ethical Statement

The study protocol was approved by the Ethical Committee (research project code: 03C023; ID: 2020_06_16_07; Acronym: CO_HABIT) of the Istituto Auxologico Italiano, Milan, Italy. All procedures performed in the study were run following the ethical standards of the institutional and/or national research committee and with the Helsinki Declaration and its later amendments or comparable ethical standards.

Written informed consent was obtained from both parents or legal guardians, and assent from adolescents was also gained.

### 2.2. Study Participants

During July 2020, 10 adolescents with obesity referring to a single inpatient clinic (Istituto Auxologico Italiano IRCCS, San Giuseppe Hospital) for a multidisciplinary body weight reduction program, entailing moderate energy restriction, adapted physical activity, psychological counseling, and nutritional education [28,29] (mean duration: 25 ± 3 days), and their primary caregivers, were consecutively recruited and screened for admission into the study.

Adolescents were selected if: (1) they were aged between 15 and 17 years; (2) used Italian as their mother tongue; (3) had a standardized Body Mass Index Standard Deviation Score (BMI SDS) ≥ 2.00 [30]; and (4) their primary caregiver(s) were 18 years of age or older. Subjects were excluded from the study if presented neurological, cognitive, and/or psychiatric problems, or hearing difficulties.

During the selection process, the gender of young patients was considered to ensure heterogeneity of the sample, while caregivers were free to decide who would participate in the research. Nine mothers and one father of an adolescent finally entered the study. The age of the adolescents ranged from 14.82 to 17.92 years (mean age = 16.98; SD = 0.92). Their mean SDS BMI was 2.43 (SD = 0.33) at inclusion and decreased by 0.17 points during the rehabilitation period on average. Caregivers were aged 44 to 59 years and their BMI ranged from 20.20 to 37.18 kg/m^2^ (mean BMI = 27.4 kg/m^2^; SD = 5.5). Eight out of ten caregivers (80%) had a high school education, whereas one had a middles school education, and one was graduated. They all had jobs: nine were employed and one caregiver was self-employed. Considering marital status, five caregivers were married/cohabiting, while the other 50% were separated or divorced. The demographic and clinical characteristics of the sample are reported in Table 1.

### 2.3. Measures

#### 2.3.1. The Lived Experience of Adolescents with Obesity and Their Caregivers

In line with the purpose of the present study, two different semi-structured interviews were created ad hoc for adolescents and their caregivers following a multi-informant approach [31]. Questions focused on specific, connected, and integrated life domains (health, education/job, recreational activities, family, and social relationships), on the individuals’ sense of self, and the lived experiences, expectations, and coping skills related to COVID-19 social isolation. A continuous measure of the level of difficulties experienced by the subjects was also provided.

The first step in developing the tool was to review existing measures of similar domains or constructs. A list of potential questions was then created and grouped into domains. Based on the collective clinical experience of all authors, items were further added to capture domains that seemed to be missing. Lastly, specific questions on the lived experience that participants made of COVID-19-related social isolation were included. All authors discussed and agreed on the final version of both measures (for adolescents and caregivers).

The final sets of included items were similar between interviews. However, the wording and response options were adapted to include developmentally appropriate scenarios and vocabulary. For example, one of the four questions composing the *expectations and coping skills* domain posits, for adolescents: “*At the end of your hospitalization, the lockdown is going to be over. How do you imagine isolation could have changed your life, your habits? … the relationships with your parents, sibling, girls, and boyfriends, and school-mates?*”. In the contrast, the caregiver version asks: “*After the lockdown, and everything seems to be slowly returning to normal, how did social isolation change—in negative and positive—your life? … the relationships with your child, friends, or colleagues.*”.

The semi-structured interviews encompassed 17 items for adolescents and 12 for caregivers which measure six primary dimensions: *health* (adolescents: four items, caregivers: four items), *education/job* (adolescents: three items, caregivers: two items), *recreational activities* (adolescents: two items, caregivers: no items), *sense of self* (adolescents: three items, caregivers: three items), *family and social relationships* (adolescents: four items, caregivers: two items), and *expectations and coping skills* (adolescents: one item, caregivers: one item). For each domain, several sub-domains were investigated. The *health* domain includes four sub-domains: objectives, motivation and self-efficacy (adolescents: four items, caregivers: four items), efficiency (adolescents: three items, caregivers: four items), stability (adolescents: three items, caregivers: three items), and satisfaction (adolescents: three items, caregivers: three items), the *education/job* domain was explored in term of perceived self-efficacy (adolescents: five items, caregivers: six items), satisfaction (adolescents: one item, caregivers: two items), and correspondence between education and professional goals (adolescents: two items, caregivers: no items). Active interest (one item) and satisfaction (one item) with *recreational activity* were investigated only in adolescents, while the domain related to the *sense of self* was considered both in adolescents and their caregiver across sub-domains of description (adolescents: one item, caregivers: one item), self-evaluation (adolescents: two to four items according to the answers given, caregivers: two items), and flexibility to change (adolescents: one item, caregivers: one item). The *family and social relationships* domain encompassed dimensions specific for family (adolescents: four items, caregivers: four items), and interpersonal relationships (adolescents: four to six items according to the answers given), description (adolescents: two items), and flexibility (adolescents: three items, caregivers: three items). Lastly, the *expectations and coping skills* dimension was specifically meant to investigate how the pandemic and consequent enforced social distancing affected participants’ lifestyle and well-being.

Considering the specific purpose of the study, questions grouped in domains were scored on a 3-point Likert scale. For each dimension, specific indications were provided to assist the interviewers in assigning the score*—*from 0 to 2*—*that most appropriately reflect the degree of difficulties/problems experienced by the subjects: 0 reflected the absence of difficulties, 2 indicated the presence of difficulties, and 1 designated an intermediate level where only some difficulties were present. A score of 3 was assigned in case the subject did not answer the question, failed*—*after repeated clarifications*—*to understand the question, or responded in an inconsistent way (none was assigned).

A total score (ranging from 0 to 34 for adolescents and from 0 to 24 for caregivers) was calculated by summing the scores for each interview domain and interpreted within the content analysis of the narratives, demographics, and clinical features of participants (detailed information and scores obtained to the semi-structured interviews by each participant are reported as Appendix A)

#### 2.3.2. Demographic and Clinical Variables

The following demographics were collected by interviewers after having commenced the semi-structured interviews: age, and gender (for both adolescents and their caregiver); education, job, and marital status (for caregivers only).

Moreover, biomedical information of weight and height were registered from both adolescents and their caregivers and used to calculate their BMI SDS [BMI-mean BMI (for age and sex)/SD] or BMI (kg/m^2^), respectively. Measures for adolescents were retrieved from their medical records, while directly asked adults.

### 2.4. Procedure

Before commencing the study, the surveys were presented to a pilot adolescent-caregiver dyad (who did not enter the analysis) to help identify potential problems with the interviews that might lead to biased answers. No major changes were needed.

Two clinical psychologists trained in qualitative interviewing (authors C.V. and M.B.) conducted the semi-structured interviews during the rehabilitation period.

The inpatient four-week hospital-based and medically managed rehabilitation program for weight reduction required that children and adolescents (age < 18 years) with obesity (a) were assessed by a staff dietician and placed on an individualized hypocaloric nutritional balanced Mediterranean style diet (80% of the Harris-Benedict estimated individual basal metabolic rate of daily caloric requirements with a composition of 59% carbohydrates, 25% fat, and 16% protein); (b) attended a nutritional education program aimed at promoting change in eating habits consisting of individual sessions and group sessions providing information on obesity and related health risks, nutrient intake recommendations, setting realistic weight loss goals, and behavior change strategies for weight management and preventing relapse; (c) engaged in physical activity once each weekday that consisted of group classes on physical therapy and aerobic activity by staff exercise physiologists; and (d) receives weekly individual and group psychological therapy focused on addressing emotional correlates of weight gain while promoting self-efficacy and problem solving, managing crises, supporting motivation and monitoring treatment progress.

School attendance was also guaranteed: junior and middle school children followed their study plan provided by teachers in the hospital, while high school students were autonomously allowed to follow the online lessons on their own devices.

#### 2.4.1. Initial Phase

Adolescents and their caregivers who met the above inclusion criteria and gave their informed consent to take part in the study were consecutively invited to participate in the study at admission to the clinic by the same psychologists responsible for administering the intervention (authors C.V. and M.B.).

On this occasion, participants were informed about the aim and procedure of the research and were asked to sign the written informed consent. With informed consent, adolescents and their caregivers also permitted professionals to audio recordings of the interview.

Individuals were also made clear that their participation was voluntary, that they would not receive any rewards for participating in the study, and that they could withdraw from the study at any stage without that affecting the rehabilitation treatment in any way—but none of them did. To ensure the anonymity of the interviewees, each adolescent-caregiver dyad was assigned a numerical identification code.

#### 2.4.2. Main Phase

The ad hoc semi-structured interviews were conducted within seven days from the beginning of the rehabilitation, were audio-recorded and lasted 39.75 min—on average—for adolescents (25.37–50.24 m) and 27.88 min for caregivers (19.30–48.00 m)

Each interview was conducted separately in order to allow patients and caregivers to freely express their opinion and perspectives [32].

Adolescents were interviewed in a dedicated room of the hospital, in privacy. Since due to COVID-19-related restrictions visits from family members and friends were not allowed during the data collection period, interviews with the primary caregivers were conducted online on a secure online platform or by phone.

The interviewers used probing questions, and active listening techniques to facilitate the expression of experiences and emotions [33,34]. Field notes and reflective notes made by the interviewers after each interview were used to prompt further reflections on the narratives [33].

#### 2.4.3. Final Phase

At the end of each interview, the professionals made sure of the participant’s physical and emotional state. Interviews were audio-recorded and transcribed verbatim. Then, the two interviewers (authors C.V. and M.B.) independently assigned the most appropriate score to each domain, and any discrepancy was resolved by a third reviewer (author V.B.). Following, narratives were subjected to qualitative analysis.

### 2.5. Intervention Fidelity

Reliability techniques employed in this study included (1) the recording and transcription of semi-structured interviews; (2) the joint formal textual analyses of narratives by two researchers; (3) the independent assignment of scores given by the two interviewers to the answers provided and the resolution of discrepancies between scores by a third researcher (author V.B.).

### 2.6. Sample Size Calculation

Although there is no standard for a minimum number of participants in qualitative research, our internal statistician experts have identified a sample size of 10 subjects as adequate to achieve saturation of the collected information [35]. Similarly, Langford and colleagues [36] and Morgan [37] recommend 6–10 individuals, and Kuzel [38] suggests that 6–8 data sources or sampling units are sufficient when selecting homogeneous samples in qualitative research. Morse [39] also suggests that qualitative researchers use at least six participants for investigations aimed to understand the essence of the experience lived by individuals.

Therefore, a sample of 10 adolescent-caregiver dyads (gender-homogeneous)—for a total of 20 subjects—was deemed sufficient for this study.

### 2.7. Analysis

Descriptive statistics were conducted to summarize the demographic and biomedical characteristics of the sample. Data from the interviews were transcribed verbatim from the audio recordings by authors C.V. and M.B., who also provided quantitative scores to each dimension.

According to the recommendation outlined by Smith and colleagues [40], transcriptions were qualitatively analyzed using an interpretative phenomenological approach (IPA)—a meticulously idiographic and hermeneutic method considered by many authors as the most effective technique to capture subjective experiences [41,42].

First, junior researchers (authors V.B. and A.G.U.) read each transcript several times to familiarize with their contents, rigorously coding line by line the text, and making memos of descriptive, linguistic, and contextual details to promptly identify interesting aspects and emerging impressions that formed the basis for emerging themes. The research team worked systematically through entire data sets, giving full and equal attention to each data item.

Meetings were also held- and recorded-throughout the coding process to allow debriefing and to help researchers to examine how their thoughts and ideas were evolving as they engaged more deeply with the data.

Then, themes were identified by bringing together components or fragments of ideas or experiences retrieved from the narratives, and, for each theme, keywords and exemplar texts were extracted.

Following, the authors looked for connections between themes: some were clustered together into master themes, while others formed subthemes. Miscellaneous themes were kept in separate free nodes to ensure they were not lost.

The themes and subthemes were not considered definitive until all of the data were analyzed by the research team and further discussed with content experts (authors G.P., R.C., and G.C.).

The themes were organized and reorganized until consensus was reached, and all team members were satisfied that all data were represented and offered in a meaningful and useful manner.

Finally, the team revisited the names of all themes and subthemes with the intent to ensure that they were representative of the content of the participants’ narratives.

A final comprehensive table of themes and subthemes was ultimately created (Table 2).

## 3. Results

The analyses of interviews revealed five master themes that characterized the life-experience of adolescents with obesity and their caregivers during quarantine: (1) COVID-19 as an opportunity to reconsider what makes a good life; (2) Persistence in Life; (3) Empowering relationship; (4) Daily routine in quarantine; (5) Lives on hold.

Each master theme comprised emergent sub-themes that were considered a further specification of the meaning of master themes. Master themes, emergent themes, and exemplificative quotations are reported in Table 2.

### 3.1. COVID-19 as an Opportunity to Reconsider What Makes a Good Life

The first theme that emerged from the interviews of both adolescents and their caregivers reflected a positive experience of lockdown. It involved noticing and being thankful for little things of everyday life they have failed to value because taken for granted. This led to greater relationship management, respect, and empathy toward others. As a result, social bonds were strengthened, and taking the time to slow down and savor life was reported as a valuable source of mental balance.

#### 3.1.1. Find Happiness in the Little Things

During quarantine, caregivers reported having the opportunity to reflect and to become more appreciative of ordinary everyday values and experiences that are usually taken for granted:

“*I realized that—before the advent of the pandemic—I tended to give many things for granted”; “in this period I have developed a deep sense of gratitude*”.(caregiver #5)

Similarly, updated information on the daily number of COVID-19 cases and deaths by country worldwide made adolescents being grateful for the ordinary everyday things.

“*I change my perspective: before COVID-19, many things were taken for granted*”.(adolescent #8)

#### 3.1.2. Make Responsible Choices

The risk for contagion made adolescents more responsible, attentive, and respectful of preventive measures. The COVID-19 global health emergency resulted in a unique chance to promote interest, responsibility, and civic engagement:

“*I am now careful not to drink from others’ bottle or borrow objects, and to wash my hands frequently*”.(adolescent #10)

#### 3.1.3. Get Pleasure from Being in Contact with Others

Beyond social distancing, COVID-19 produced emotional closeness between people. Caregivers reported the importance of living in contact with others and of mutual-aid components:

“*Cooperation and living in empathy with others are a strength because it is easier to ask for help and -in turn—to assist to someone*”.(caregiver #2)

### 3.2. Persistence in Life

Throughout interviews, adolescents and caregivers pointed out that—contrary to expectations—the pandemic did not dramatically change who they are and the society where they live. They defined themselves as the same people they were before the COVID-19 outbreak. At the same time, they were confident that life should return to normal soon.

#### 3.2.1. Enduring Self-Representation

Adolescents and caregivers perceived themselves as the same persons they were before the COVID-19 epidemic and believed that the spread of the virus did change dramatically their intimate nature and self-representations.

#### 3.2.2. Return to Normality

Most adolescents and caregivers did not doubt that their lives will return as they were before the COVID-19 outbreak and were confident that society will change drastically as a consequence of the virus:

“*I will always remember that I lived the COVID-19 pandemic, but I think that we will get back to our normal routine very soon*”.(adolescent #1)

### 3.3. Empowering Relationship

Another recurrent theme reported by adolescents and their caregivers concerned the domain of interpersonal relationships. Some respondents affirmed that spending more time at home, and sharing home space during quarantine, had a positive impact on family relationships. Cooking, watching television, and dialoguing were some of the main shared activities reported by the dyads. Still, increased conflicts between family members during the lockdown were also reported by some teenagers. Moreover, friendships and social contact were recognized as strengthened by both adolescents and their caregivers.

#### 3.3.1. Together at Home

Spending a lot of time together at home during the quarantine resulted in an enjoyable experience for most of the interviewed. Both adolescent and their caregivers admitted that it was unexpected and had a positive influence on their family relationships:

“*Being together has been positive for our relationship*”.(caregiver #3)

Particularly, adolescents learned to appreciate the importance of sharing time while collaborating with their parents and siblings in housework and stated that this helped them to develop a greater positive sense of self-worth while improving their domestic skills.

Still, forced cohabitation also led—in some cases—to an increase in conflict and disagreements between family members:

“*We fought more! Staying at home…what a nightmare*!”(adolescent #8)

#### 3.3.2. Strengthened Social Relationship

Narratives revealed that the lockdown contributed to bolstering relationships with schoolmates/colleagues and friends in both adolescents and their caregivers via online tools and social networking sites.

“*I feel in contact with my friends*”.(adolescent #2)

“*With my colleagues, there was a greater union: we gave each other advice and support, we talked about our worries …*”.(caregiver #4)

### 3.4. Daily Routine in Quarantine

Assessment of the effects of COVID-19-induced confinement policies on nutritional habits and adherence to physical activity recommendations led to different outcomes in adolescents. Adjustments to the new conditions were particularly relevant for those living with a parent affected by COVID-19.

#### 3.4.1. Adherence to Recommended Lifestyle

During COVID-19 adherence to healthy habits was—for some adolescents—facilitated in different ways including the impossibility to buy palatable foods and the presence of a supporting parenting style.
“*During the pandemic, dieting was easier to follow because of the lockdown. I was not allowed to go out and buy my favorite foods*”.(adolescent #2)
“*She (daughter) correctly followed the recommendations (diet and physical activity) because I was at home: I cooked for her and she exercised with her sister*”.(caregiver #7)

On the contrary, other respondents reported an increase in sedentary lifestyles and unhealthy food consumption.
“I *used to seat all day, playing videogames or attending online school lessons, while continuously eating regardless of whether it was breakfast lunch or dinner time*”(adolescent #7)
*“I used to spend more than 23 h doing nothing, I just felt the need to eat, no matter what. I tried to control myself, but I have been regularly tempted*”.(adolescent 5#)
“*He (son) couldn’t go out, to the gym…this was the greatest limit that the lockdown posted on adhering to prescriptions. He spent all day in his room*”.(caregiver #9)

Emotional eating also emerged in response to experienced stressful thoughts and feelings:
*“I coped with my stressful moments by eating*”.(adolescent #7)

#### 3.4.2. Welcoming COVID-19 at Home

Specific themes emerged in youth whose parents were affected by COVID-19 or worked in hospital settings. This not only had an impact on the physical closeness between family members but also on the quality of their interpersonal relationships, as well as on individual welfare and responsibility for household tasks.
“*She (mother) was self-confined in her bedroom, and I had to deal with my schoolwork, at the same time tidying up home. My brothers did not help me at all, and we often argued for this reason*”.(adolescent #2)
“We *had several rules for when I returned from work. For example, we separated the use of everything including towels and home-spaces. My children ate sitting on the table in the living-room while I used to stay in the kitchen. We wore the medical masks all the time… it was dramatic*”.(caregiver #2)

### 3.5. Lives on Hold

High levels of uncertainty for the future were also found in some adolescents and their caregivers, with the consequent perpetuation of controlling behaviors in the attempt to regain a sense of normality.

#### 3.5.1. Living with Uncertainly

Due to the lack of clear and precise information and regulations by the government, the future seemed uncertain to most of the interviewed. This strongly limited the possibility for caregivers to plan for future actions and make long-term decisions.
“*I made fewer plans for the future. I live more in the here and now…I do not know if it will be possible to plan something for next year*”.(caregiver #7)

#### 3.5.2. The Missed Routine

Interviewed believed that restoration of the previous situation was a distant goal, in some cases unattainable—and associated feelings of hopelessness also emerged:
“*I fear the normality won’t come back. We can’t do anything about it*”.(caregiver #2)
“*It seems that everyone is afraid… normality seems far away*”.(adolescent #5)

#### 3.5.3. Take Control

The strict preventive measures imposed by the government to flatten the curve of the pandemic also led individuals to experience fears and anxiety, which young people and adults have tried to face by implementing different control behaviors.
“*I’m very careful. I use only FFP2 masks and I pay attention to every single activity in my everyday life*”. (caregiver #7)
“*Whatever I do, I wash my hands*”.(adolescent #2)

## 4. Discussion

The increasing number of young people with obesity makes it significant the examination of the impact of quarantine on their physical and mental health, and the study of the role of the family environment in moderating the effects of social isolation and COVID-19 on their mood and behavior.

To our knowledge, this is the first study that employed an interpretative phenomenological methodology to detail the perspectives and life experiences of adolescents living with obesity and their caregiver during the pandemic.

The experiences described by participants provided a valuable insight that raised important considerations for improvement in support services.

Themes emerging from our data suggest that social isolation was surprisingly reported as a valuable source of mental balance, since taking the time to slow down and savor life enabled participants to reflect on the meaning and of what is important in life. This led to greater relationship management, respect, and empathy toward others.

Adolescents in this study highlighted the importance of and their need for support during quarantine from their families and peers, and assistance and encouragement allowed them to discuss sensitive issues, provided stability in their lives, and protected them from unpleasant experiences.

Further, sharing time and responsibility for self-care and housework between caregivers and adolescents led to increased self-efficacy, psychological health, and self-care behaviors among respondents in this study.

Accordingly, previous research show that parents’ abilities to trust their children’s self-management behaviors positively impact the quality of their relationships and on their behavioral outcomes [43,44,45,46,47].

On the contrary, when communication between family members was poor or conflicting, or caregivers exerted too much control over the eating and exercise habits of the adolescents, data from this study revealed emerging emotional issues in adolescents that affected their self-management abilities and adaptation to the novel situation.

Exploring the dyadic experience of adolescents with obesity and their primary caregivers also served as a unique opportunity to detect discrepancies between narratives. For instance, while adherence to the recommended lifestyle of her daughter was reported as satisfactory by the primary caregiver, the adolescent reports it as dramatically poor. This further emphasizes the importance of effective communication between family members to allow a better understanding of reciprocal needs and concerns and the provision of a supportive buffering presence to youth.

Living with a caregiver affected by COVID-19 or working in the hospital setting also impinged upon adolescent autonomy, as the risk for contagion and accompanying high levels of parental stress further increased the difficulty for caregivers to offer timely assistance and encouragement to their children. It also required young respondents to take full responsibility for domestic tasks and schoolwork, with a further detrimental effect on family cohesion and intimacy.

Undeniably, the family environment plays a key role in facilitating a supportive environment for adolescents with obesity during colliding pandemics.

In this study, a parenting style characterized by unnecessary control over adolescents’ behaviors, as well as excessive permissive and neglectful educational approaches negatively affected the ability of young people to self-regulate. Results support the adoption of a responsive but demanding authoritative parenting style [48], which provides the care needed for adolescents to autonomously fosters self-control and responsibility for their behaviors, including compliance with the preventive measures imposed by the government to reduce the spread of the virus.

In agreement, several cross-sectional studies have found an association between authoritative parenting style and lower BMI in youth [49,50,51,52,53], and one longitudinal study found that children of authoritarian parents were five times more likely to be overweight or obese [54].

Due to the lack of clear information by the government and perceived limited chances for caregivers to make a long-term plan for the future, perpetration of controlling behaviors in the attempt to regain a sense of normalcy paradoxically led to increased uncertainty in both adolescents and their caregivers. While reflecting on the current living situation made them largely recognize the benefits of social isolation, when asked about the future helplessness as a function of fear of failure to return to normality arise in both adolescents and their caregivers.

Interesting, none or limited emotional difficulties were detected by the scores of the interviewees who believed that—once the pandemic will be over—everything will return as before. In contrast, higher difficulties—mostly in the sense of self dimension—were found in adolescents and caregivers who considered returning to normal to be impossible, and to perceive no control over the situation. The concept of locus of control refers to the amount of control a person believes to exert on specific events in their life [55]—and not surprisingly strongly correlates with the constructs of autonomy and self-efficacy.

Some authors have noted that an external locus generally leads to emotional difficulties, which further affects the individuals’ ability, motivation, and confidence in dealing with tasks of daily-life (e.g., weight-management) [56]. Internally focused people with obesity would, therefore, tend to succeed more often in weight-loss endeavors than their externally focused counterparts [57,58,59]. Notice, the direction of causality is very likely reversed, with obesity causing a more external locus of control—particularly in youth [60].

Interventions specifically targeted to develop an internal locus of control in individuals with obesity—and to manage expectations accordingly—would help straightening their resources and self-efficacy by creating healthier habits and intentions.

The family environment also shaped the adolescents’ eating habits through a direct influence on their nutrition and physical effort. In fact, caregivers have demonstrated role models for eating behaviors [61], and parental obesity is a key determinant for the onset of overweight problems during childhood and adolescence [62,63]. Six out of the ten caregivers interviewed in this study felt in the overweight and obese weight-range—with two of them reporting a BMI over 35 kg/m^2^. A correspondence between higher weight in caregivers and their children was also observed.

Interestingly, these adolescents also seemed to suffer more the consequences of social isolation compared to those with a primary caregiver in the normal weight-range—particularly in the recreational activity life-domains. Consequently, they seemed to face greater challenges in remaining hopeful about the potential for a better life.

Research on youth supports the strong impact of perceived self-efficacy on the participation in leisure activities of youth with obesity [64,65]—and theories of motivation and behavioral change postulate that the best approach to increase youth participation in recreational and physical activity is to enhance their enjoyment by progressively improving their perceived skill competence [65,66,67,68]. In addition, the study by Crossman et al. (2006) concluded that adolescents who have less time to engage in sedentary activities because they are involved in any other activities—including non-athletic activities—such as part-time jobs, volunteer work, or household chores are less likely to develop overweight or obese as young adults [69]. Indeed, lack of effective time management and boredom were other determinants of poor eating practices and exercise reported by the adolescents interviewed in this study as consequences of prolonged time at home.

Adolescence is inherently a time of development and change, where family and peer relationships have a key influence on physical and mental health [70].

Online platforms and social media were extensively used to communicate with friends before the COVID-19 pandemic, with a demonstrated increase of sedentary habits.

Still, virtual learning and consequent higher screen time during social isolation prevented some of the adolescents in this study from relapsing on prohibited palatable foods, and respondents also reported a significant increase in the quality of their relationships with friends, schoolmates, and colleagues [71,72].

Notably, adolescents living with obesity experience unfavorable comparisons with other adolescents while at school [73,74] due to the predominant aesthetic models of our society [75,76].

School closures and the use of digital technology during quarantine may, therefore, have served as a protective factor against potential exposures to weight-stigma by offering respondents a safe environment for disclosure of personal feelings and experiences in selected interactions with peers.

Stigma impacts on shame and self-criticism, contribute to the development of a negative perception of oneself as worthless and inadequate to others [77], and it is widely associated with undermining self-regulation of eating behavior, body image dissatisfaction [78,79,80,81] binge eating [82,83] and obesity [84].

Participants in this study suggested that being in a group with other adolescents suffering from obesity (both virtually and during the ongoing rehabilitation period) was a preferred arena to share experiences, obtain peer support and perhaps participate in pleasant activities with others.

### 4.1. Strengths and Limitations

A major feature and benefit of this study lie in its rigorous qualitative methodology to explore a novel area of research encompassing obesity and COVID-19 pandemics.

Applying an interpretive phenomenological approach to the collection and analysis of data allowed the exploration of deeper meanings and phenomena related to the impact that social isolation had on the mood and behaviors of adolescents with obesity and their primary caregivers at a moment in time. Moreover, to properly meet the aim of the study, the present research focuses on the dyadic experience that adolescents and their caregivers made of social isolation during the COVID-19 pandemic.

No self-report measures of mood and behavior were included, and this might represent a concern because they could add to the effects of social isolation on lifestyle behavior. However, quantitative data were also used to improve *credibility* and *confirmability* through critical reflection, and to reduce interpretation bias between researchers. Demographic characteristics were also explored in relation to the data to increase the probability that the research findings and interpretations will be found credible.

The credibility of the analysis was further enhanced by having two researchers analyze each data set, both with proven experience in working with this particular population of patients and their caregivers.

However, due to feasibility considerations, while adolescents were interviewed in person, semi-structured interviews with caregivers were conducted online or by telephone, thus altering the credibility of the study results. Furthermore, responses might have been richer if fewer questions were asked to identify family dynamics that might affect weight management in adolescents with obesity. Other weaknesses might be the challenges that the respondents faced in terms of putting their experiences and worries into words, and social desirability bias.

Still, a collaborative analysis with the abstraction of data in reflexive dialogues and several steps were performed, challenging pre-understandings and interpretations in constant dialogues, and increasing credibility of research findings.

The research process is logical, traceable, and documented, but since the duration of participant observation and overall engagement in this study was short-term, lack of d*ependability* might reduce the overall trustworthiness of the present results and more research is warranted to capture additional features of adolescent-caregiver dyads and weight-management practices over a longitudinal period.

Participants were highly representative of the population under study. However, recruitment took place in a single rehabilitation center. This relative homogeneity of the sample may affect *transferability* since regional and cultural differences were underestimated. Further research in other settings (nationally and internationally) would be beneficial to broaden the understanding of the relationships between the emerging relevant interacting components affecting weight-management in adolescents.

### 4.2. Future Research and Practical Directions

This study provides a valuable insight into the lives and experiences of adolescents with obesity and their primary caregiver during social isolation and offers practical implications for helping adolescents with obesity maintain a healthy weight. Social, and behavioral pathways to obesity within the family environment, and perceived quality of family relationships are worthy of investigation in the early steps of the rehabilitation. Parents pass on values and norms to their children by communicating their views and selectively reinforcing or discouraging behaviors.

To determine if the parents’ controlling behavior is the cause or the outcome of their children’s excessive weight would also have important implications for clinical intervention aimed at promoting a supportive and empowering parenting style. Positive feedback mechanisms would reinforce safe and successful participation in recreational and physical activities and would increase perceived self-esteem. Given the evidence for the reciprocal relationship between lower self-esteem and excessive weight, its prompt evaluation is essential to deliver interventions able to improve functioning, weight loss, and quality of life in adolescents.

Exploring the perspective of adolescents and their caregivers on the problem also increases the therapeutic alliance between health care providers and family members. Families play an active role in shaping their children’s transition plan. As healthcare providers focus on the comprehensive care of children and their families, a systematic assessment of their health and biopsychosocial needs is critical to the reduction of the negative impact of obesity and COVID-19.

## 5. Conclusions

Curbing the viral spread while protecting population health will remain a top priority until an effective COVID-19 vaccine is available or even longer. However, it is imperative to address other co-existing problems such as the impact that social isolation has on emotional and behavioral features of adolescents with obesity, which may have long-term profound health and economic consequences than the actual COVID-19 infection.

## Figures and Tables

**Table 1 ijerph-18-03026-t001:** Demographic and clinical characteristics of the sample.

Variables	Adolescents	Caregivers
	Mean (SD)|range	
Age (years)	16.98 (0.92)|14.82–17.92	50.00 (4.50)|44–59
Weight_pre	98.8 (15.05)|79–124	
Weight_post	95 (13.92)|76–118	
SDS BMI_pre	2.42 (0.33)|1.86–3.01	
SDS BMI_post	2.25 (0.32)|1.71–2.72	
BMI (kg/m^2^)		27.4 (5.5)|20.20–37.18
	*n* (%)	
Education		
Elementary school diploma		0 (0)
Middle school diploma		1 (5)
High school diploma		8 (80)
University Degree		1 (10)
Job position		
Employed		9 (90)
Self-employed		1 (10)
Marital status		
Single		1 (10)
Married		4 (40)
Separated or divorced		5 (50)

**Table 2 ijerph-18-03026-t002:** Master themes, emergent themes, and quotations of the interviews.

Master Themes	Emergent Themes	Quotations
COVID-19 as an opportunity to reconsider what makes a good life	Find happiness in the little things	“*I realized that—before the advent of the pandemic—I tended to give many things for granted*” *(caregiver #5).* “*I now live my life more peacefully: today we are here, tomorrow we are not here anymore. Why bother?*” *(caregiver #1).* “*The increasing number of deaths both in Italy and worldwide makes you think*” *(adolescent #6).* “*We enjoy more each day; we give them more value*” *(adolescent #7).* “*I change my perspective: before COVID, many things were taken for granted*” *(adolescent #8).*
	Make responsible choices	“*I am now careful not to drink from others’ bottle or borrow objects, and to wash my hands frequently*” *(adolescents #10).* “*We have to learn about what is happened in the past*” *(adolescent #6).*
	Get pleasure from being in contact with others	“*Cooperation and living in empathy with others are a strength because it is easier to ask for help and -in turn—to assist someone*” *(caregiver #2).*
Persistence in life	Enduring self-representation	“*I am the same person as always*” *(adolescents #4).* “*I think I will be the same person as before*” *(caregiver #9).*
	Back to normality	“*I will always remember that I lived the COVID-19 pandemic, but I think that we will get back to our normal routine very soon*” *(adolescent #1).* “*I don’t think the pandemic will change many things … we should probably be more careful at the beginning, but nothing more*” *(caregiver #10).*
Empowering relationship	Together at home	“*Before the COVID-19 outbreak we were all detached in family, but thanks to the lockdown, we had the opportunity to stay together at home and talk about many different things*” *(adolescent #2).* “*Being together has been positive for our relationship*” *(caregiver #3).* “*For the first time I did my laundry, I washed dishes, I dried clothes, so I learned a lot of things*” *(adolescent #7).* “*We fought more! Staying at home…what a nightmare!*” *(adolescent #8).*
	Strengthened social relationship	“*I feel in contact with my friends*” *(adolescent #2).* “*Relations with my colleagues have improved: we gave each other advice and support, we talked about our worries …*” *(caregiver #4).*
Daily routine in quarantine	Adherence to recommended lifestyle	“*During the pandemic, dieting was easier to follow because of the lockdown. I was not allowed to go out and buy my favorite foods*” *(adolescent #2).* “*She (daughter) correctly followed the recommendations (diet and physical activity) because I was at home: I cooked for her and she exercised with her sister*” *(caregiver #7).* “*I used to seat all day, playing videogames or attending online school lessons, while continuously eating regardless of whether it was breakfast lunch or dinner time* “ *(adolescent #7).* “*I used to spend more than 23 h doing nothing, I just felt the need to eat, no matter what. I tried to control myself, but I have been regularly tempted*“ *(adolescent 5#).* “*He (son) couldn’t go out, to the gym…this was the greatest limit that the lockdown posted on adhering to prescriptions. He spent all day in his room*” *(caregiver #9).* “*I coped with my stressful moments by eating*” *(adolescent #7).*
	Welcoming COVID-19 at home	“*She (mother) was self-confined in her bedroom, and I had to deal with my schoolwork, at the same time tidying up home. My brothers did not help me at all, and we often argued for this reason*” *(adolescent #2).* “*We had several rules for when I returned from work. For example, we separated the use of everything including towels and home-spaces. My children ate sitting on the table in the living-room while I used to stay in the kitchen. We wore the medical masks all the time… it was dramatic*” *(caregiver #2).*
Lives on hold	Living with uncertainly	“*I made fewer plans for the future. I live more in the here and now…I do not know if it will be possible to plan something for next year*” *(caregiver #7).*
	The missed routine	“*I fear the normality won’t come back. We can’t do anything about it*” *(caregiver #2).* “*It seems that everyone is afraid… normality seems far away*” *(adolescent #5).*
	Take control	“*I’m very careful. I use only FFP2 masks and I pay attention to every single activity in my everyday life*” *(caregiver #7).* “*Whatever I do, I wash my hands. I feel them dirty, so I wash them again*” *(adolescent #2).* “*I hope, in the future, I won’t feel fear and distrust when I stay with others*” *(caregiver #9).*

Notes: Quotations were translated from Italian.

## Data Availability

The data presented in this study are available on request from author G.P. with the permission of author A.S. The data are fully reported in Italian and are not publicly available due to privacy/ethical restrictions.

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
