# Peer review of "The Impact of Social Isolation during the COVID-19 Pandemic on Physical and Mental Health: The Lived Experience of Adolescents with Obesity and Their Caregivers"

_ijerph, 2021, doi:10.3390/ijerph18063026_

Round 1

Reviewer 1 Report

This manuscript is an original and novel study aimed to identify perspectives and patterns in the lived-experiences of a sample of adolescents with obesity and their caregiver during the COVID-19 pandemic. Background could be enriched with more evidence to express smoothly and a well logic flow for readers to catch up, instead of using too many separate small paragraphs to explicit ideas. Research design is appropriate and methods are described in sufficient detail. Results part adequately encompasses transcribed data by using comprehensive tables and explanations of each theme. However, “Results” are a little lengthy for reading, because the quotations with many long sentences of each theme appeared twice, taking up a lot of space, which could be presented shortly. Significance and practical directions in the future is explicitly detailed in the “Discussion”. Additionally, results are concluded sufficiently and previous research findings are incorporated to fully discuss the significance of this study and close the research gap. However, the same suggestion as above “Introduction” should be mentioned, discussion could be refined and reorganized to be more concise and avoid using many separate paragraphs with a few sentences.

Author Response

Dear reviewer, thank you for reviewing our manuscript and for your helpful suggestions, we incorporated them accordingly and answered step by step including all the suggestions of the reviewers and the answers respectively in the cover letter, kind regards, Henriette Löffler-Stastka

Reviewer 2 Report

Please see attached

Author Response

(The authors gave the same response as above.)

Reviewer 3 Report

This is an interesting piece of work which has been thoroughly conducted. I have a few questions/suggestions aimed at enhancing clarity rather than anything else. 

Are you sure you want to call it a phenomenological hermeneutic  approach? I am not convinced it really is, in the discussion you call it interpretative phenomenology which seems more accurate. If it is it may warrant a greater level of explanation/rationalisation. 

Line 71 nasty parental practices seems judgemental and harsh is that from reference 22?

Putting the long list  domains and subdomains in table rather than prose may assist with clarity.

Why were there only 3 points on the Likert scale, warrants some rationalisation? 

If a score for 3 was given for them not understanding the question, was that added to their final score, needs a little explanation or tidy 

ref 29 in line 126 is in red, not sure why. 

column heading needs to be at the top of each page for the results 

Author Response

(The authors gave the same response as above.)

Round 2

Reviewer 1 Report

i am satisfied with the revised version and would suggest for publication.